

# There is no magic bullet: the importance of testing reference gene stability in RT-qPCR experiments across multiple closely related species

Bert Foquet and Hojun Song

Department of Entomology, Texas A&M University, College Station, TX, United States of America

## ABSTRACT

Reverse Transcriptase quantitative Polymerase Chain Reaction (RT-qPCR) is the current gold standard tool for the study of gene expression. This technique is highly dependent on the validation of reference genes, which exhibit stable expression levels among experimental conditions. Often, reference genes are assumed to be stable a priori without a rigorous test of gene stability. However, such an oversight can easily lead to misinterpreting expression levels of target genes if the references genes are in fact not stable across experimental conditions. Even though most gene expression studies focus on just one species, comparative studies of gene expression among closely related species can be very informative from an evolutionary perspective. In our study, we have attempted to find stable reference genes for four closely related species of grasshoppers (Orthoptera: Acrididae) that together exhibit a spectrum of density-dependent phenotypic plasticity. Gene stability was assessed for eight reference genes in two tissues, two experimental conditions and all four species. We observed clear differences in the stability ranking of these reference genes, both between tissues and between species. Additionally, the choice of reference genes clearly influenced the results of a gene expression experiment. We offer suggestions for the use of reference genes in further studies using these four species, which should be taken as a cautionary tale for future studies involving RT-qPCR in a comparative framework.

## INTRODUCTION

The current gold standard tool for studying gene expression at the RNA level is Reverse Transcriptase quantitative Polymerase Chain Reaction (RT-qPCR or simply qPCR), due to its high sensitivity and speed of analysis (*Gachon, Mingam & Charrier, 2004*; *Thellin et al., 2009*). Nonetheless, qPCR accuracy is highly dependent upon the normalization of target gene expression with reference genes. An optimal reference gene should show minimal variability in its expression level between tissues, be unaffected by tested experimental factors, and exhibit similar expression levels as target genes (*Vandesompele et al., 2002*). Often those genes that are critical for maintaining basic cellular functions and expressed in all cell types, commonly referred to as housekeeping genes, are used as reference genes

Corresponding author
Bert Foquet, bertfoquet@tamu.edu

for qPCR experiments. Although qPCR is one of the basic tools employed in functional genetic research, mistakes in the experimental setup for qPCR experiments are surprisingly common, including the use of an inappropriate number of reference genes or the lack of accurate testing of reference gene performance under specific experimental conditions (*Bustin et al., 2013*; *Gutierrez et al., 2008*; *Kozera & Rapacz, 2013*). It is often the case that certain reference genes are selected for a particular qPCR experiment simply because they have been used previously, either for other experimental conditions or even in other tissues and species. This type of blind adoption of reference genes can result in inaccurate normalization of target gene expression, and ultimately in an incorrect interpretation of the results (*Bustin et al., 2013*; *Dheda et al., 2004*; *Gutierrez et al., 2008*; *Nicot et al., 2005*; *Tricarico et al., 2002*; *Vandesompele et al., 2002*). In response to these issues, the Minimum Information for Publication of Quantitative Real-Time PCR Experiments (MIQE)-guidelines, aiming to enhance the consistency of performing, interpreting, and reporting qPCR data, were formulated (*Bustin et al., 2009*). Additionally, several statistical algorithms have been developed to identify the best reference genes to use under certain experimental conditions (*Andersen, Jensen & Orntoft, 2004*; *Hellemans et al., 2007*; *Pfaffl et al., 2004*; *Vandesompele et al., 2002*). Therefore, it is critical that a thorough investigation of reference gene stability is needed prior to setting up any qPCR experiment.

While a vast majority of qPCR-based studies focus on gene expressions on a single species, there is a recognition that a comparative gene expression study across multiple closely related species that may have distinct biological or ecological differences can reveal important insights into the evolution of gene functions (e.g., *Salazar-Jaramillo et al., 2017*; *Sørensen et al., 2019*; *Wittkopp, Vaccaro & Carroll, 2002*). However, one of the assumptions in this type of comparative studies is that the reference genes that work well for one species must also work well for another closely related species. Although the reference genes are supposed to be stable within a species (*Vandesompele et al., 2002*), there is no a priori reason to believe that the same pattern is found in another species. If this assumption does not hold true, the subsequent inferences about the expression level of the gene of interest can be incorrect. Nevertheless, this assumption is rarely tested.

In this study, we explore the merit of this assumption by testing reference gene stability in qPCR experiments in four closely related species of grasshoppers in the genus *Schistocerca* (Orthoptera: Acrididae). Our initial motivation for this work comes from our long-term interest in understanding the molecular basis of density-dependent phenotypic plasticity in locusts. In short, locusts are grasshoppers that show an extreme form of density-dependent phenotypic plasticity in which relatively inactive and solitary individuals can transform into highly active and gregarious individuals in response to change in local population density (*Cullen et al., 2017*; *Pener, 1983*; *Pener & Simpson, 2009*; *Uvarov, 1921*). When the high density persists, locusts exhibit collective movements, which can lead to locust plagues (*Pener & Simpson, 2009*). These two density-dependent phenotypes are called solitarious and gregarious phases (*Pener & Simpson, 2009*; *Uvarov, 1921*), and understanding the molecular basis of this phenomenon has been considered the last frontier in locust research (*Cullen et al., 2017*; *Pener & Simpson, 2009*). We have been studying the Central American locust, *Schistocerca piceifrons* (Walker), an important swarming locust species affecting

Mexico and Central America (*Barrientos Lozano et al, 1992*; *Bredo, 1963*; *Harvey, 1983*), as a model system, which shows behavioral, morphological, physiological, ecological, and molecular plasticity in response to change in density, similar to its more well-studied congener, the desert locust *S. gregaria* (Forskål). Our research has shown that the Central American locust is more closely related to non-swarming grasshoppers than to other locust species within the genus (*Song et al., 2017*), and that these non-swarming relatives also show density-dependent phenotypic plasticity, reminiscent of the swarming locusts (*Gotham & Song, 2013*; *Song et al., 2017*). Therefore, over the past several years, we have been investigating the evolution of density-dependent phenotypic plasticity in a comparative framework by comparing transcriptomes of *S. piceifrons* and three other closely related *Schistocerca* species, which have led to the identification of some candidate genes that might be relevant for the expression of density-dependent phenotypes. It is in this context that we ask a question about the suitability of using the same reference genes for qPCR experiments across the four closely related species. In this study, we have designed and analyzed the stability of eight potential reference genes across the four species reared in two density conditions (isolated vs. crowded), and we demonstrate that the assumption of reference gene stability is not supported in our study system. We also validate our choice of reference genes with a set of four candidate genes in *S. piceifrons* by performing qPCR experiments. Finally, we provide a set of recommendations for selecting appropriate reference genes in a comparative analysis involving multiple closely related species.

## MATERIALS & METHODS

### Study insects and rearing regime

We used four closely related species in the genus *Schistocerca*, maintained as laboratory colonies in the Department of Entomology at Texas A&M University. The four species were the Central American locust, *S. piceifrons*, and three sedentary species, *S. americana* (Drury), *S. serialis cubense* (Saussure), and *S. nitens* (Thunberg). For conciseness, we use the species epithet (*piceifrons*, *americana*, *cubense*, and *nitens*) throughout the paper. The *piceifrons* colony was established from an outbreak population in Yucatan, Mexico collected in October 2015, and imported under a USDA permit (USDA APHIS PPQ P526P-15-03851). The *americana* colony was established from a population in Brooksville, Florida, collected in September 2010. The *cubense* colony was established from a population in Islamorada in the Florida Keys collected in January 2011. Finally, the *nitens* colony was established from a population in Terlingua, Texas, collected in May, 2015. For this study, the colonies of these four species were reared under crowded and isolated conditions. For the isolated condition, nymphs were isolated as hatchlings and reared in separate plastic cages ($10.16 \times 10.16 \times 25.4$ cm) with one transparent side and connected to a filtered positive airflow. For the crowded condition, both cage size and the number of specimens depended on the species. For *piceifrons*, about 200 nymphs were kept in a large cage ($40.64 \times 34.29 \times 52.07$ cm) in a USDA approved quarantine facility. For *americana* and *cubense*, 150–200 nymphs were kept in larger cages ($30.48 \times 35.56 \times 50.8$ cm). For *nitens*, over 50 individuals were kept together in a small cage ($30.48 \times 35.56 \times 50.8$ cm) because

we observed that the insects would die in the density used for other species. In both density conditions, the insects were reared at 12 h of light and 12 h of darkness at 30 °C, and were fed daily Romaine lettuce and wheat bran. We reared the insects until they molted to the last nymphal instars to conduct qPCR experiments.

## RNA extraction and cDNA synthesis

Sample collection, RNA extraction and RNA quality assessment were performed as previously described (*Wang et al., 2020*). Briefly, crowded-reared and isolated-reared female nymphs were marked with a ceramic marker on the abdomen after molting to their last nymphal instar, and were dissected around 72 h later. Only specimens that molted before 10 AM were used, and all dissections occurred between 8 and 9 AM. After removing gut tissues, head and thorax were dissected using sterilized dissection scissors. Both tissues were preserved in RNAlater (ThermoFisher Scientific) at −20 °C, following the manufacturer's guideline. A total of 10 nymphs/density/species were dissected. Half of these was used for RNA sequencing, and the other half was used for the qPCR experiment. Tissues were placed in MagNA Lyser Green Beads (Roche) sample tubes and were homogenized for 30 s in one mL of Trizol (ThermoFisher Scientific) using a MagNa Lyser instrument (Roche) at 6,500 rounds per minute. Whole-tissue RNA was extracted using a Trizol-chloroform extraction, followed by clean-up with a RNeasy mini kit (Qiagen) using an on-column DNAse treatment with a RNase-free DNAse set (Qiagen). RNA concentrations were measured with a Denovix DS-11 spectrophotometer; 260/280 and 260/230 values were above 2.0 for all samples. Additionally, microcapillary electrophoresis with a Fragment Analyzer (Agilent Technologies RNA) was used to analyze RNA integrity of samples destined for RNA sequencing. Only those samples with a RNA Quality Number (RQN) over 3.9 were used. Due to differences in 28S ribosomal RNA structure compared to other eukaryotic species, RQN values for insects are often lower than what is generally considered a valid threshold in mammalian samples (*Escobar & Hunt, 2017*; *Macharia, Ombura & Aroko, 2015*; *Winnebeck, Millar & Warman, 2010*). Samples used for qPCR were diluted to a concentration of 100 ng*ml$^{-1}$ and subsequently used to synthesize cDNA using the iScript cDNA synthesis kit (Bio-Rad) following the manufacturer's guideline.

## RNA sequencing and transcriptome assembly

Both RNA sequencing and transcriptome assembly used for the present study were previously described in detail in *Wang et al. (2020)*. Briefly, we generated RNA-seq data by performing paired-end sequencing (150 bp) on 8.5 lanes on an Illumina HiSeq4000 (San Diego, CA). Library preparation, sequencing, and read formatting was performed at Texas A&M's AgriLife Research Genomics and Bioinformatics Service. After initial processing of raw data, raw reads were imported into a personalized Galaxy environment (*Afgan et al., 2018*) on a supercomputing cluster of the High-Performance Research Computing group of Texas A&M University (Ada, https://hprc.tamu.edu) for further processing, trimming, and quality control. FastQ Screen (*Wingett & Andrews, 2018*) was used to filter out sequences from the following potential contamination sources: (UniVec core (June 6, 2015), PhiX (NC_001422.1), Illumina adapters, *Gregarina niphandrodes* genome

(GNI3), *Encephalitozoon romaleae* genome (ASM28003v2), *Escherichia coli* genome (K12), *Methylobacterium* sp., *Bosea* sp., *Bradyrhizobium* sp., *Klebsiella pneumoniae*, *Sphingomonas* sp., *Rhodopseudomonas* sp. and *Propionibacterium acnes*). We used Trinity (*Grabherr et al., 2011*) for de novo assembly, which was performed separately for each species and tissue. We further filtered the resulting assemblies using CD-hit-EST (*Fu et al., 2012*; *Li & Godzik, 2006*) and Transrate (*Smith-Unna et al., 2016*). We assessed transcriptome quality using Trinitystats (*Grabherr et al., 2011*), BUSCO (Benchmarking Universal Single-Copy Orthologs; (*Simão et al., 2015*), and by calculating the fraction of reads mapping back to the transcriptome with a combination of bowtie2 (*Langmead & Salzberg, 2012*; *Langmead et al., 2009*) and flagstat (*Li, 2011a*; *Li, 2011b*; *Li et al., 2009*).

## Sequence assembly and expression analysis

All eight transcriptomes were imported into Geneious (R10.2.6; BioMatters, Ltd.). We selected nine potential reference genes: actin5C (*Act5C*), α-tubulin (*Tub*), succinate dehydrogenase (*SDH*), elongation factor 2 (*EF2*), ribosomal protein L5 (*RIBL5*), glyceraldehyde-3-phosphate dehydrogenase (*GAPDH*), annexin IX (*Ann*), armadillo (*Arm*) and heat shock protein 70 (*Hsp70*). We also selected four target genes: pacifastin-related peptide 4, pacifastin-related peptide 5, allatostatin (*ast*), and allatotropin (*at*). For consistency, we followed suggestions by *Simonet et al. (2004)* for naming pacifastin-related peptides and named the *Schistocerca piceifrons* sequences *SPPP-4* and *SPPP-5*. We obtained nucleotide sequences for all 13 genes from other orthopteran insects from Genbank (https://www.ncbi.nlm.nih.gov/genbank/) and used Megablast or tblastx with default settings in Geneious to find related sequences in our four species. For each gene, sequences of all four species were aligned using MUSCLE version 3.8.24 (*Edgar, 2004*) in Geneious, with a maximum of nine iterations and default settings. Subsequently, sequences were manually curated, resulting in full-length coding sequences for each species. The identity of each gene was confirmed using the standard nucleotide Basic Local Alignment Search Tool (blastn) at the NCBI website, using the nr/nt database as reference (*Altschul et al., 1990*; *Johnson et al., 2008*). Sequence information can be found in Table 1. Additionally, mapping reads were counted as previously described (*Wang et al., 2020*), using bowtie2 (very sensitive end-to-end, disable no-mixed and no-discordant behavior) and SAMtools' idxstats (*Li et al., 2009*) in Galaxy. Differential expression analysis was performed with DEseq2 (*Love, Huber & Anders, 2014*) within SARTools 1.7.1 (*Varet et al., 2016*), using R 3.5.3 (*R Core Team, 2017*). All settings were kept at their defaults; the expression in crowded-reared individuals was compared to that of conspecific isolated-reared individuals.

## Primer design

Primers for reference genes (Table 2) were designed using Primer3 (*Koressaar & Remm, 2007*; *Untergasser et al., 2012*), based on conserved regions, identified using the nucleotide alignment in Geneious. In this way, we aimed to generate primers that would work in all four species. Standard settings were altered to an optimal melting temperature set at 60 °C, and the amplicon length at 150–200 bp. For the four target genes, primers were

**Table 1** Reference genes and target genes used in this study and their known functions, with Genbank accession numbers.

| Gene | Full name | Associated Genbank accession number | | | | Function |
|------|-----------|------------|-----------|---------|--------|----------|
| | | *piceifrons* | *americana* | *cubense* | *nitens* | |
| Act5C | actin 5C | MT498271 | MT498272 | MT498273 | MT498274 | Structural constituent of cytoskeleton |
| Tub | α-tubulin | MT498316 | MT498313 | MT498314 | MT498315 | Structural constituent of cytoskeleton |
| Hsp70 | heat shock protein 70 | MT498304 | MT498301 | MT498302 | MT498303 | Stress response, protein folding |
| RIBL5 | ribosomal protein L5 | MT498312 | MT498310 | MT498311 | MT498309 | Structural component of ribosome, protein translation |
| EF2 | elongation factor 2 | MT498294 | MT498295 | MT498296 | MT498293 | Protein synthesis |
| GAPDH | glyceraldehyde phosphate dehydrogenase | MT498297 | MT498299 | MT498300 | MT498298 | Oxidoreductase in glycolysis & gluco-neogenesis |
| Ann | annexin IX | MT498288 | MT498285 | MT498286 | MT498287 | Formation of membrane scaffolds, actin binding |
| Arm | armadillo | MT498292 | MT498289 | MT498290 | MT498291 | Wnt signal transduction pathway |
| ast | allatostatin | MT498275 | | | | Pleiotropic neuropeptide, downregulation of JH production |
| at | allatotropin | MT498280 | | | | Pleiotropic neuropeptide, upregulation of JH production |
| SPPP-4 | pacifastin-related peptide 4 | MT498307 | | | | Peptidase inhibitor |
| SPPP-5 | pacifastin-related peptide 5 | MT498308 | | | | Peptidase inhibitor |

designed for just *piceifrons*, with identical settings in Primer3 as described above (Table 2). We designed three primer pairs for each gene. All sequences were ordered from Integrated DNA Technologies.

## Real time quantitative PCR and statistics

For each qPCR reaction, 5 μl of cDNA was added to 10 μl of SsoAdvanced[TM] Universal SYBR[®] Green Supermix (Bio-Rad #1725275) and 5 μl of primers at a final concentration of 500 nM. Every reaction was performed in duplicate or triplicate. To determine primer efficiency for reference genes, a dilution series of one sample was generated for each species by performing serial 10-fold dilutions ranging from dilutions of 1/1 to 1/10,000. For the target genes, a 5-fold dilution series from 1/1 to 1/3,125 was used instead for determining primer efficiency. Based on these data, the most efficient primer pair was used in all further analyses. All other reactions were run on 96-well plates, with 10 samples (five isolated-reared individuals and five crowded-reared individuals) run on the same plate for a particular gene following the sample maximization method (*Hellemans et al., 2007*). Reactions were run on a CFX connect real time system (Bio-Rad) using the following thermal cycling profile: 3 min at 95 °C, 40 cycles of (1) 15 s at 95 °C and (2) 45 s at 60 °C, and a melting curve from 65 °C to 95 °C. $C_q$ values were exported from the Bio-Rad CFX manager using the default threshold.

**Table 2  Primer information for qPCR experiment.** Primer sequences, amplicon melt temperature and primer efficiencies are given. Primer efficiencies were obtained with a 10-fold dilution series ranging from 1/1 to 1/1,000,000.

| Gene code | Primer sequences | piceifrons | | americana | | cubense | | nitens | |
|---|---|---|---|---|---|---|---|---|---|
| | | Tm (°C) | E (%) | Tm (°C) | E (%) | Tm (°C) | E (%) | Tm (°C) | E (%) |
| Act5C | F: AACTTTCAACACCCCAGCCA R: AACGCCATCACCAGAATCCA | 82.5 | 102.13 | 82.5 | 105.84 | 82.3 | 97.65 | 82.0 | 104.21 |
| Tub | F: AGCTCATCACTGGCAAGGAG R: TCCTGATGCGATCCAACACC | 81.5 | 103.15 | 81.5 | 100.30 | 81.5 | 104.90 | 82.0 | 96.18 |
| Hsp70 | F: TCGTCAACTCAAGCCAGCAT R: TGCTTTCTCCACAGGTTCCA | 80.5 | 100.18 | 80.3 | 105.06 | 80.3 | 102.16 | 80.0 | 104.14 |
| RIBL5 | F: TCGGCTGCACAGAAGTTACC R: AGCTCCAGTAGTTGTGCGGA | 83.7 | 99.83 | 83.7 | 101.20 | 83.5 | 96.78 | 84.0 | 102.11 |
| EF2 | F: CATCTCCTGTGGTTGCACAG R: ATGACACCAACTGCTGCTTC | 79.3 | 100.00 | 79.0 | 102.73 | 79.3 | 101.33 | 79.3 | 98.98 |
| GAPDH | F: TGGCTTTCAGAGTCCCAGTG R: AGCAGCTTCCTTCACCTTGG | 84.0 | 102.28 | 84.0 | 97.74 | 83.7 | 98.57 | 83.3 | 98.53 |
| Ann | F: ATAGGGGAATTGTGCAGCGG R: TGCCACTCAGTTCACTCTTC | 79.7 | 100.10 | 79.7 | 102.39 | 79.5 | 99.47 | 79.5 | 104.14 |
| Arm | F: TCTCAGAGAGTTCGTGCTGC R: GGCTTGGTTCTGCTAGACGT | 82.3 | 109.85 | 82.7 | 101.41 | 82.5 | 99.27 | 82.5 | 92.12 |
| ast | F: TCAACCAAACCCGCCTCAAG R: ACACTACCCCACAGAGAAGC | 81.3 | 96.84 | | | | | | |
| at | F: GATGCAGAACAACCCGGAAC R: CAGTAAGTGGGCCTGAGGAG | 85.0 | 91.26 | | | | | | |
| SPPP-4 | F: ACTCCAGGAACCATGAAGAAGG R: AGGAGTGCAGTTTACCTCTCTC | 83.5 | 91.19 | | | | | | |
| SPPP-5 | F: AGCTGTACACCCAACTCGAC R: TCGTTCCAGGAGTGCAGTTC | 87.0 | 94.51 | | | | | | |

**Notes.**

Tm, melt temperature of the PCR amplicon; E, primer efficiency.

For the analysis of reference gene stability, a total of five crowded-reared and five isolated-reared individuals per species were used for both head and thorax tissues. Gene stability was analyzed using three different programs: geNorm in qbase+ (*Mestdagh et al., 2009*; *Vandesompele et al., 2002*), NormFinder (*Andersen, Jensen & Orntoft, 2004*), and BestKeeper (*Pfaffl et al., 2004*). Raw $C_q$ values were used as input for BestKeeper and geNorm, while they were transformed to a linear scale for NormFinder. Gene efficiencies were assumed to be 100% for all tested genes (see Table 2 for actual gene efficiencies). To obtain a non-arbitrary ranking of reference genes, they were first ranked for each program separately, and subsequently the geometric mean of these scores was calculated as an accurate estimate of which reference genes were the most stable (*Chen et al., 2011*; *Chen et al., 2013*; *Gong et al., 2016*; *Pereira-Fantini et al., 2016*). The amount of reference genes needed to standardize gene expression was assessed using geNorm. Normalization factors were calculated in a stepwise manner, starting by taking the two most stable reference genes into account and adding another reference gene one by one. Subsequently, the variation between these normalization factors were compared. If the difference in variation is lower

than 0.15, the addition of an additional reference gene was deemed unnecessary (*Mestdagh et al., 2009*; *Vandesompele et al., 2002*).

For the quantification of target gene expression, we used thorax tissues of five isolated-reared and five crowded-reared individuals of *piceifrons*. Relative expression of target genes (*ast, at, SPPP-4,* and *SPPP-5*) for thorax tissue in *piceifrons* was calculated with the $\Delta\Delta C_q$-method (*Livak & Schmittgen, 2001*), using different sets of reference genes to normalize gene expression. *P*-values were calculated with a two-tailed student *t*-test in R (version 3.6.2) based on non-transformed $\Delta C_q$-values. All raw qPCR data can be found in Data S1–S3, and the code for the student *t*-test is present in Data S1.

## RESULTS

### Selection of potential reference genes

A total of 9 potential reference genes were initially selected based on previous studies using qPCR for locust gene expression research (*Chapuis et al., 2011*; *Van Hiel et al., 2009*; *Yang et al., 2014*). All of these genes were members of different functional categories (Table 1), with the exception of the cytoskeleton genes, *Tub* and *Act5C*, which are both commonly used as reference genes in insect gene expression studies (*Lü et al., 2018*). We chose *EF2* and *RIBL5* over elongation factor 1α and other ribosomal genes, respectively, based on the low amount of variation for these genes in our transcriptome data. For 7 out of 9 genes, primer efficiencies for the selected primer pairs varied between 96 and 106% (Table 2). Two exceptions were *Arm* with an efficiency of 109.8% for *piceifrons* and 92.1% for *nitens*, and *SDH* for which we were unable to find primers with sufficient efficiency values in all four species. We attribute this to a surprisingly high amount of sequence differences among the four species for *SDH*, effectively restricting the sequence regions that were conserved enough for primer design. As a result, *SDH* was removed from all further analyses, and thus, only eight reference genes were used in further analyses. All included primer pairs showed melt curves with a single peak, suggesting the absence of aspecific amplification and contamination. $C_q$ values varied between 17.5 and 28.5, with *GAPDH* and *Hsp70* having the lowest values and *Arm* having the highest.

### Comparing reference gene stability

We subsequently compared the stability of these eight potential reference genes by performing qPCR reactions in five isolated-reared and five crowded-reared individuals for head and thorax tissues in all four species (Fig. 1). In *americana*, *cubense*, and *nitens*, all eight genes showed similar expression levels under the tested rearing conditions. However, in *piceifrons*, several genes showed a trend towards differential expression between the two density conditions. Specifically, *Ann, Act5C, Tub*, and *Hsp70* showed a trend towards lower expression in the isolated condition for both tissues, while *Arm* and *EF2* showed this trend only in the thorax tissue. *RIBL5* was the only gene showing no difference between the two density conditions. We were able to confirm most of these observed trends with our transcriptome data (Table 3). The only discrepancies between our transcriptome data and the qPCR data were found for *Hsp70* in head tissue, *GAPDH* for thorax tissue, and
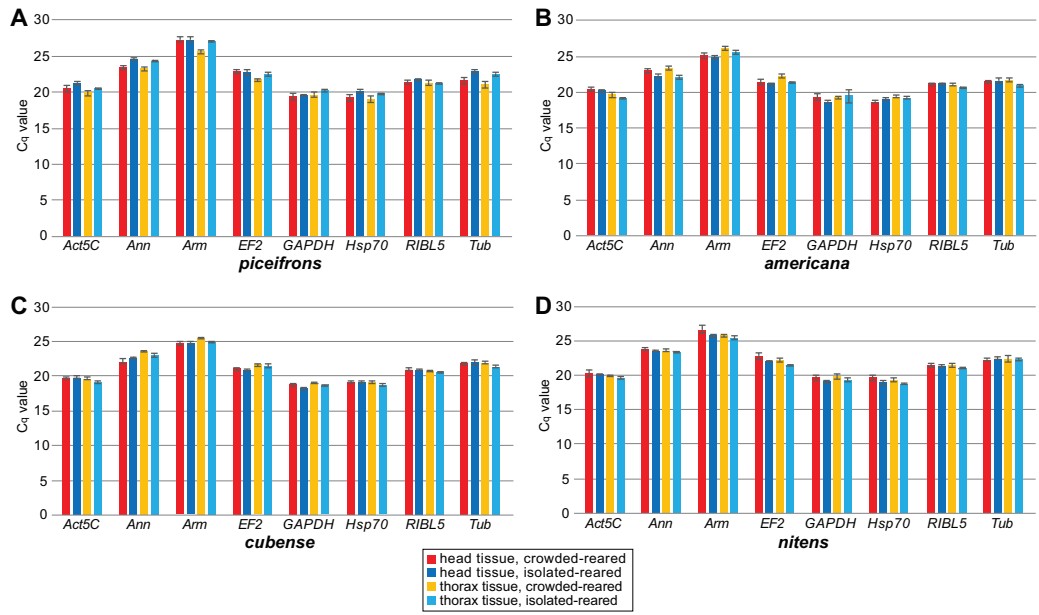

**Figure 1** **Mean C_q values of reference genes for qPCR study.** $C_q$ values were obtained by qPCR for all tested reference genes for all tissue, rearing condition and species combinations. Boxes and error bars represent the mean $\pm$ standard error of mean of $C_q$ values. Each group contained five biological replicates. Graphs represent observations for (A) *piceifrons*, (B) *americana*, (C) *cubense* and (D) *nitens*.

**Table 3** **Fold changes of reference genes in piceifrons obtained from RNA-Seq experiment.** Fold changes were calculated by Deseq2 in SARTools. The isolated-reared condition was taken as reference. Five isolated-reared and five crowded-reared last instar nymphs were used. Adjusted *p*-values were obtained using default settings.

| Gene | Head | | Thorax | |
|------|------|------|--------|------|
| | Fold change | Adjusted *p*-value | Fold change | Adjusted *p*-value |
| *Act5c* | 1.255 | 0.865 | 1.602 | 0.037 |
| *Ann* | 1.420 | 0.815 | 1.654 | 0.033 |
| *Arm* | 1.081 | 0.973 | 1.270 | 0.293 |
| *EF2* | 0.983 | 0.997 | 1.189 | 0.598 |
| *GAPDH* | 1.050 | 0.988 | 0.995 | 0.993 |
| *Hsp70* | 0.966 | 0.993 | 1.201 | 0.501 |
| *RIBL5* | 0.910 | 0.972 | 1.134 | 0.796 |
| *Tub* | 0.878 | 0.954 | 0.897 | 0.832 |

*Tub* for both tissues. For thorax tissue, *Act5C* and *Ann* were even found to be significantly upregulated in crowded-reared individuals compared to isolated-reared individuals.

NormFinder (*Andersen, Jensen & Orntoft, 2004*) calculates both intra- and inter-group variations of gene expression, and combines these into a stability value, and thus, genes with a lower rank are considered to be more stably expressed. geNorm (*Mestdagh et al., 2009*; *Vandesompele et al., 2002*), the most popular algorithm, assumes that expression values of real reference genes are perfectly correlated with each other in all tested samples. Based on

this correlation, genes get a stability value (M value), with genes with $M < 1.5$ considered to be stable and the most stable gene obtaining the lowest score. BestKeeper (*Pfaffl et al., 2004*), on the other hand, outputs the standard deviation for each reference gene, expected to be lower than 1, and a correlation coefficient for each gene with the so-called BestKeeper-index, which is essentially the geometric mean of all stable reference genes. Before comparing the stability values, we estimated the amount of necessary reference genes for each species and tissue using geNorm's V-values. For *americana*, *cubense*, and *nitens*, geNorm suggested that using two reference genes was sufficient as its values for $V_{2/3}$ were below 0.15 (Fig. 2). However, for *piceifrons*, the difference in variation between using two or three reference genes was higher than 0.15 for the head tissue and very close to 0.15 for the thorax tissue (Fig. 2). Thus, the use of three reference genes is suggested for this species. Overall, the three programs listed genes in a similar fashion, with only a few exceptions (Tables 4 and 5). *Tub* and *GAPDH* were the two least stable genes, and both were listed as the last in 3 out of the 8 tested species/tissue combinations. *Act5C* be the most stable reference gene overall, as it was generally listed in the top half of stability values by all three programs. *Arm* was also often listed in the top three of most stable genes, but was also listed as the seventh most stable gene in some cases (Normfinder, Bestkeeper in *piceifrons* head; Normfinder in *nitens* head).

For the *americana* head tissue, *Act5C* and *EF2* were the two most stable genes overall, with *Act5C* rated the most stable gene by each program, and *EF2* obtaining a second place for both geNorm and NormFinder. *RIBL5* and *Arm,* which were also consistently ranked as stable reference genes, obtained a third and a fourth place. For the *americana* thorax tissue, *RIBL5* and *EF2* were our two reference genes of choice, as they obtained a first and a second place overall. Additionally, *Arm* and *Act5C* were ranked as third and fourth overall, but obtained very similar stability scores to EF2 and were ranked in the first half by all three programs. For the *cubense* head tissue, both *Act5C* and *Hsp70* were ranked within the top 3 most stable reference genes by all three programs and as a result, obtained top overall rankings. For the *cubense* thorax tissue, *Hsp70* obtained the best ranking, followed by *Ann* ranked second. Nonetheless, it should be noted that for this species and tissue combination, several reference genes obtained very similar scores for both NormFinder and geNorm, resulting in low rankings for some genes that were actually quite stable. For instance, *Act5C* was ranked as only fifth overall even though it performed very similar to *Ann*, which was ranked second. For the *nitens* head tissue, the rankings suggested by the three different programs disagreed strongly with each other, as *Ann* and *Act5C* were ranked first and second by geNorm and NormFinder, but fifth and fourth by BestKeeper, while *Arm* was ranked first by BestKeeper and sixth and seventh by the other two programs. Overall, *Ann* and *Act5C* were ranked first and second, and *Hsp70*, which was the only gene ranked within the first half by all three programs, was ranked as third. For the *nitens* thorax tissue, *Ann* and *Arm* were clearly the preferred reference genes as they were consistently ranked as the two most stable genes by all three programs. Finally, for both tissue of *piceifrons*, *Act5C* and *Ann* were ranked in the top 4 of most stable genes by all programs. *RIBL5* and *Arm* were also ranked in the top 4 for head and thorax tissue respectively.

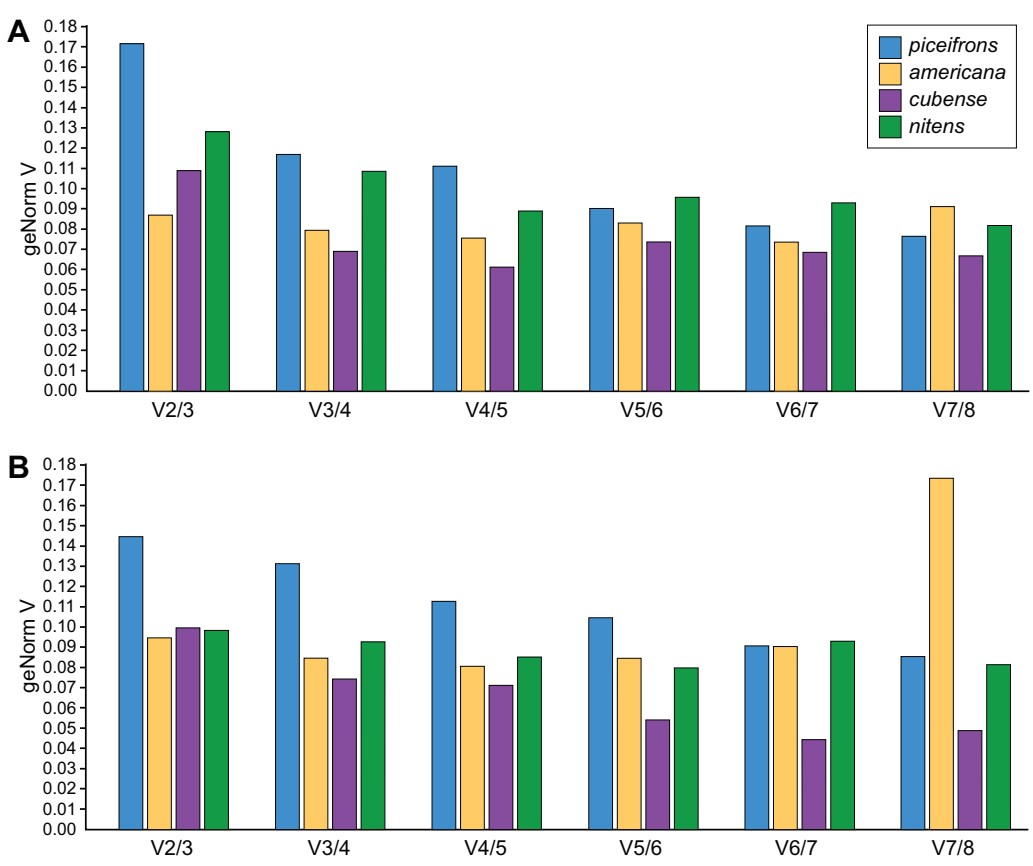

**Figure 2 Pairwise variation analysis for qPCR reference genes.** All data were generated in qbase+ using the geNorm algorithm, based on five biological replicates per group. Values represent the pairwise variation analysis between normalization factors $NF_n$ and $NF_{n+1}$, or the decrease in variation when adding an additional reference gene. Using 0.15 as a threshold, the optimal number of reference genes required for accurate normalization can be determined. Graphs are shown for (A) head and (B) thorax tissues for each species.

## Validation of reference genes

To explore how the choice of reference genes would affect the outcome of qPCR experiments, we quantified and compared the expression levels of the four target genes (*ast, at, SPPP-4,* and *SPPP-5*) using different reference genes in the thorax tissue of *piceifrons* reared in two density conditions. As *Tub* was the least stable reference gene, it was not included for this analysis. Our data showed the choice of reference genes strongly affected the qPCR results (Fig. 3). Regardless of the reference genes used, *ast* showed a significantly higher expression level in isolated-reared individuals compared to crowded-reared individuals, and *SPPP-5* never came close to reaching significant differences. When all of the reference genes (except *Tub*) were chosen, *at* and *SPPP-4* showed trends toward decreased and increased expression, respectively, in crowded-reared individuals ($p = 0.068$ and $p = 0.061$, respectively). We subsequently removed *RIBL5* from the analysis and thus only used reference genes of which the expression varied between both rearing conditions. As a result, the $p$-value for *at* dropped below 0.05, while the trend observed for *SPPP-4*

**Table 4  Stability values for reference genes in *piceifrons* and *americana*.** Stability values for eight reference genes were obtained using three different algorithms: geNorm, NormFinder and BestKeeper. Stability values and ranking from most stable to least stable are given for each gene. A comprehensive ranking was calculated by taking the geometric mean of rankings obtained by the different algorithms. Head and thorax tissue were investigated separately. For each group, five isolated-reared and five crowded-reared last instar nymphs were used.

| Species/Tissue | Gene | geNorm | | NormFinder | | BestKeeper | | | Comprehensive ranking |
|---|---|---|---|---|---|---|---|---|---|
| | | Rank | M | Rank | Stability value | Rank | std. dev. | r | |
| | Act5C | 1 | 0.506 | 1 | 0.117 | 1 | 0.630 | 0.993 | 1 |
| | Ann | 3 | 0.579 | 3 | 0.265 | 5 | 0.576 | 0.894 | 3 |
| | Arm | 6 | 0.669 | 7 | 0.341 | 7 | 0.581 | 0.742 | 8 |
| *piceifrons* head | EF2 | 4 | 0.648 | 6 | 0.333 | 6 | 0.597 | 0.761 | 6 |
| | GAPDH | 7 | 0.706 | 5 | 0.303 | 8 | 0.463 | 0.725 | 7 |
| | Hsp70 | 5 | 0.668 | 4 | 0.295 | 4 | 0.710 | 0.895 | 4 |
| | RIBL5 | 2 | 0.562 | 2 | 0.207 | 3 | 0.427 | 0.911 | 2 |
| | Tub | 8 | 0.719 | 8 | 0.375 | 2 | 0.855 | 0.931 | 5 |
| | Act5C | 2 | 0.560 | 2 | 0.148 | 4 | 0.533 | 0.925 | 2 |
| | Ann | 1 | 0.500 | 1 | 0.131 | 1 | 0.615 | 0.972 | 1 |
| | Arm | 3 | 0.592 | 4 | 0.241 | 2 | 0.747 | 0.941 | 3 |
| *piceifrons* thorax | EF2 | 4 | 0.671 | 5 | 0.251 | 7 | 0.444 | 0.676 | 5 |
| | GAPDH | 6 | 0.749 | 6 | 0.262 | 6 | 0.527 | 0.694 | 7 |
| | Hsp70 | 5 | 0.720 | 3 | 0.219 | 5 | 0.696 | 0.851 | 4 |
| | RIBL5 | 8 | 0.782 | 8 | 0.405 | 8 | 0.337 | 0.477 | 8 |
| | Tub | 7 | 0.757 | 7 | 0.344 | 3 | 0.865 | 0.937 | 6 |
| | Act5C | 1 | 0.402 | 1 | 0.053 | 1 | 0.383 | 0.944 | 1 |
| | Ann | 5 | 0.525 | 6 | 0.223 | 3 | 0.499 | 0.868 | 5 |
| | Arm | 4 | 0.497 | 4 | 0.166 | 2 | 0.525 | 0.906 | 3 |
| *americana* head | EF2 | 2 | 0.450 | 2 | 0.136 | 4 | 0.438 | 0.856 | 2 |
| | GAPDH | 8 | 0.788 | 8 | 0.280 | 7 | 0.569 | 0.556 | 8 |
| | Hsp70 | 6 | 0.579 | 7 | 0.263 | 8 | 0.328 | 0.553 | 7 |
| | RIBL5 | 3 | 0.456 | 3 | 0.158 | 5 | 0.289 | 0.826 | 4 |
| | Tub | 7 | 0.613 | 5 | 0.223 | 6 | 0.411 | 0.715 | 6 |
| | Act5C | 2 | 0.578 | 3 | 0.174 | 4 | 0.358 | 0.820 | 4 |
| | Ann | 7 | 0.799 | 7 | 0.250 | 5 | 0.679 | 0.784 | 7 |
| | Arm | 4 | 0.627 | 2 | 0.111 | 2 | 0.559 | 0.850 | 3 |
| *americana* thorax | EF2 | 3 | 0.601 | 4 | 0.176 | 1 | 0.485 | 0.853 | 2 |
| | GAPDH | 8 | 1.435 | 8 | 0.432 | 8 | 0.855 | 0.416 | 8 |
| | Hsp70 | 5 | 0.634 | 6 | 0.189 | 7 | 0.287 | 0.522 | 6 |
| | RIBL5 | 1 | 0.548 | 1 | 0.098 | 3 | 0.284 | 0.837 | 1 |
| | Tub | 6 | 0.712 | 5 | 0.182 | 6 | 0.566 | 0.700 | 5 |

**Notes.**
M, stability value calculated by geNorm; Std. dev., standard deviation on $C_q$ values of reference gene; r, correlation coefficient of reference gene with BestKeeper-index.

partially disappeared. Lastly, we included just *Hsp70*, *RIBL5* and *GAPDH* as reference genes, which was our preferred set of reference genes based on the earlier data. As a result, *SPPP-4* showed a significantly higher expression level in the crowded-reared condition, while no significant difference was observed for *at* anymore ($p = 0.19$). Three out of the four of these qPCR results agreed with our transcriptome data (Table 6).

**Table 5  Stability values for reference genes in *cubense* and *nitens*.** Stability values for eight reference genes were obtained using three different algorithms: geNorm, NormFinder and BestKeeper. Stability values and ranking from most stable to least stable are given for each gene. A comprehensive ranking was calculated by taking the geometric mean of rankings obtained by the different algorithms. Head and thorax tissue were investigated separately. For each group, five isolated-reared and five crowded-reared last instar nymphs were used.

| Species/Tissue | Gene | geNorm Rank | geNorm M | NormFinder Rank | NormFinder Stability value | BestKeeper Rank | BestKeeper std. dev. | BestKeeper r | Comprehensive ranking |
|---|---|---|---|---|---|---|---|---|---|
| | Act5C | 1 | 0.385 | 2 | 0.097 | 2 | 0.332 | 0.896 | 1 |
| | Ann | 6 | 0.540 | 7 | 0.225 | 7 | 0.457 | 0.707 | 7 |
| | Arm | 5 | 0.421 | 3 | 0.111 | 1 | 0.415 | 0.905 | 3 |
| cubense head | EF2 | 3 | 0.410 | 5 | 0.145 | 5 | 0.279 | 0.815 | 4 |
| | GAPDH | 8 | 0.590 | 8 | 0.238 | 8 | 0.418 | 0.539 | 8 |
| | Hsp70 | 2 | 0.400 | 1 | 0.095 | 3 | 0.305 | 0.872 | 2 |
| | RIBL5 | 4 | 0.413 | 4 | 0.118 | 6 | 0.263 | 0.733 | 5 |
| | Tub | 7 | 0.548 | 6 | 0.169 | 4 | 0.542 | 0.841 | 6 |
| | Act5C | 6 | 0.390 | 5 | 0.101 | 3 | 0.390 | 0.840 | 5 |
| | Ann | 4 | 0.383 | 2 | 0.091 | 2 | 0.414 | 0.861 | 2 |
| | Arm | 2 | 0.368 | 4 | 0.098 | 4 | 0.360 | 0.814 | 3 |
| cubense thorax | EF2 | 7 | 0.397 | 7 | 0.111 | 5 | 0.384 | 0.788 | 7 |
| | GAPDH | 3 | 0.378 | 3 | 0.097 | 7 | 0.263 | 0.721 | 4 |
| | Hsp70 | 1 | 0.350 | 1 | 0.075 | 1 | 0.337 | 0.893 | 1 |
| | RIBL5 | 5 | 0.387 | 6 | 0.110 | 8 | 0.273 | 0.708 | 6 |
| | Tub | 8 | 0.458 | 8 | 0.129 | 6 | 0.406 | 0.752 | 8 |
| | Act5C | 2 | 0.529 | 2 | 0.158 | 4 | 0.453 | 0.916 | 2 |
| | Ann | 1 | 0.528 | 1 | 0.139 | 5 | 0.282 | 0.867 | 1 |
| | Arm | 6 | 0.714 | 7 | 0.248 | 1 | 0.601 | 0.935 | 4 |
| nitens head | EF2 | 5 | 0.621 | 6 | 0.214 | 2 | 0.646 | 0.933 | 5 |
| | GAPDH | 7 | 0.721 | 5 | 0.199 | 7 | 0.482 | 0.584 | 7 |
| | Hsp70 | 4 | 0.578 | 3 | 0.178 | 3 | 0.539 | 0.924 | 3 |
| | RIBL5 | 3 | 0.567 | 4 | 0.198 | 6 | 0.439 | 0.792 | 6 |
| | Tub | 8 | 0.749 | 8 | 0.277 | 8 | 0.266 | 0.074 | 8 |
| | Act5C | 4 | 0.517 | 3 | 0.101 | 5 | 0.281 | 0.778 | 4 |
| | Ann | 1 | 0.448 | 1 | 0.020 | 1 | 0.270 | 0.934 | 1 |
| | Arm | 3 | 0.499 | 2 | 0.057 | 2 | 0.433 | 0.868 | 2 |
| nitens thorax | EF2 | 6 | 0.656 | 5 | 0.157 | 6 | 0.526 | 0.727 | 6 |
| | GAPDH | 7 | 0.710 | 4 | 0.135 | 7 | 0.595 | 0.703 | 7 |
| | Hsp70 | 5 | 0.549 | 6 | 0.164 | 3 | 0.449 | 0.866 | 5 |
| | RIBL5 | 2 | 0.471 | 7 | 0.170 | 4 | 0.300 | 0.819 | 3 |
| | Tub | 8 | 0.736 | 8 | 1.628 | 8 | 0.610 | 0.691 | 8 |

**Notes.**
M, stability value calculated by geNorm; Std. dev., standard deviation on $C_q$ values of reference gene.; r, correlation coefficient of reference gene with BestKeeper-index.

## DISCUSSION

The overall goal of this study is to address the suitability of using the same reference genes for qPCR experiments across multiple closely related species. So far, only a handful of studies have compared reference gene stability between different species (e.g., *Axtner & Sommer, 2009*; *Bonnet et al., 2013*; *Giménez, Pistón & Atienza, 2011*; *Lin et al., 2009*; *Weyrich, Axtner*

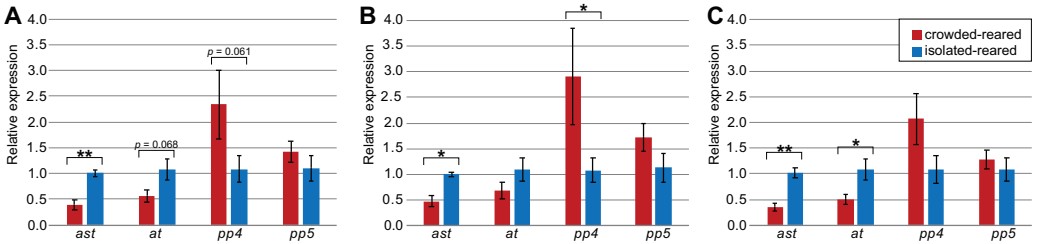

**Figure 3** **Influence of selected reference genes on target gene expression in qPCR.** Relative concentrations were calculated with the $\Delta\Delta C_q$-method, with values shown in the graph representing the exponentialized values. Error bars represent the standard error of mean, calculated on exponentialized values. $p$-values were calculated based on the $\Delta C_q$ values. All data is based on the expression in five isolated-reared and five crowded-reared individuals. Significant differences between isolated-reared and crowded-reared individuals were marked for $p < 0.05$ (*) or $p < 0.01$ (**). (A) All available reference genes were used; *Act5C, Ann, Arm, Hsp70, EF2, GAPDH* and *RIBL5*, (B) only suggested reference genes were used; *RIBL5, GAPDH* and *Hsp70*, (C) All reference genes showing a trend towards differential expression in thorax were used; every reference gene except *RIBL5*.

**Table 6** **Comparison of target gene expression using RNA-Seq and using RT-qPCR.** Fold changes for RNA sequencing study were obtained with Deseq2 in SARTools, adjusted pvalues were obtained using default settings. Fold changes for RT-qPCR study were obtained using $\Delta\Delta C_q$ method, $p$-values were obtained with a student $t$-test in R. For each analysis, five isolated-reared and five crowded-reared last instar nymphs were used. For both analyses, the isolated-reared condition was used as reference.

| Gene | RNA sequencing | | RT-qPCR | |
|------|------|------|------|------|
| | Fold change | Adjusted *p*-value | Fold change | *p*-value |
| *ast* | 0.83 | 0.865 | 0.48 | 0.018 |
| *at* | 1.06 | 0.974 | 0.63 | 0.195 |
| *SPPP-4* | 2.09 | 0.029 | 2.68 | 0.032 |
| *SPPP-5* | 0.69 | 0.713 | 1.53 | 0.156 |

*& Sommer, 2010*). Many qPCR-based studies have simply used the reference genes designed and optimized for one species in a study involving another species without specifically testing for their gene stability (*Bustin et al., 2013*; *Lin et al., 2009*; *Lü et al., 2018*). In this study, we have compared the stability of eight potential reference genes across four *Schistocerca* species reared in two density conditions, and demonstrated that the reference gene stability cannot be assumed *a priori* in a comparative analysis across multiple species.

The choice of reference genes used in qPCR experiments is often guided by the genes used in prior studies, rather than by a comprehensive assessment of all available candidates. *Lü et al. (2018)* performed a meta-analysis on the selection of reference genes in insect qPCR experiments based on 90 papers published between 2013-2017, and showed that the same 10 reference genes were repeatedly used across 78 insects belonging to 10 different orders. There are certainly more than 10 reference genes available and we think that there could be hundreds or even thousands of potential reference genes in any given species. For example, there are over 3,800 genes that are expressed uniformly across tissues in humans (*Eisenberg & Levanon, 2013*), and hundreds of reference genes have been utilized

in human gene expression studies. The reason why only a small number of reference genes are used in insect gene expression studies is probably because we do not fully understand gene contents and their expression patterns in most insects. Therefore, when designing qPCR experiments for non-model insect species, one has no other option but to rely on the previous studies to identify well-known reference genes. Our study was no exception. We selected our reference genes based on studies performed in other locust species, including the desert locust *Schistocerca gregaria* (*Van Hiel et al., 2009*), the migratory locust *Locusta migratoria* (*Yang et al., 2014*), and the Australian plague locust *Chortoicetes terminifera* (*Chapuis et al., 2011*). However, only one of these (*Chapuis et al., 2011*) included rearing density as an experimental factor, and found *Arm* and *Elongation Factor 1α* as the most stable reference genes, while *GAPDH* and *SDH* were found to be the least stable. Even though we did not test exactly the same genes, *GAPDH* is overall also one of the least stable reference genes, while A*rm* is often found within the top 3 of most stable genes in our study. This suggests that, to a certain degree, stable reference genes in one species can be expected to be stable in a related species, even though there is a clear need for scrutiny as the reference gene stability varied widely across both species and tissues in our study.

We have designed primers for eight reference genes that can be used across four species of *Schistocerca*. In doing so, we observed an interesting trend when comparing primer efficiencies between different species. As the four *Schistocerca* species used in our study are closely related, we designed and tested a single primer pair per each gene that would work for all four species. We consistently selected conserved gene regions for the primer design, and made sure that the selected regions did not overlap with sites showing single nucleotide polymorphisms. Nonetheless, the primer efficiencies for some primer pairs varied widely (Table 2). As such, it seems that a base pair difference in the amplicon, or maybe even upstream or downstream of it, can influence the primer efficiency. This indicates that primer efficiencies should always be tested for every species (or even subspecies/populations) included in a comparative study, even if those species are very closely related to each other.

After designing the primers for the reference genes and testing their efficiencies, we assayed their gene stability using three commonly used programs (NormFinder, geNorm, and BestKeeper). Interestingly, we found that different programs selected different genes as their top choices, and the results also depended on both tissue types and species (Tables 4 and 5). This discrepancy is probably due to the different algorithms employed by these programs.

We have also identified additional issues when these reference genes are used for a species exhibiting an extreme form of phenotypic plasticity. Of the four species analyzed in this study, *piceifrons* is the only swarming locust species, and can be expected to exhibit larger molecular differences between both density conditions than the other three species. It is thus feasible that several of the tested reference genes could also be influenced by the extreme form of density-dependent phenotypic plasticity. In fact, we report that several reference genes in *piceifrons* showed a trend towards differential expression. Additionally, geNorm predicted that the use of two reference genes would not be sufficient to perform a qPCR experiment for *piceifrons*, in contrast to what the program suggested for the
other three species. All of the samples were given exactly the same treatment independent of rearing condition and were all diluted to 100 ng/μl before the cDNA synthesis. As such, we consider it unlikely that all isolated-reared samples somehow had lower RNA concentrations than the crowded-reared samples. Given the extreme form of density-dependent phenotypic plasticity in *piceifrons*, it is conceivable that some of the tested reference genes actually respond differentially to the rearing conditions. This idea was supported by our transcriptome data, where most genes exhibited a similar trend towards differential expression as found in our qPCR, and two genes even were significantly different between both rearing conditions (Table 3). These observations make the selection of suitable reference genes for *piceifrons* a lot more complicated compared to that for the other three species. To make the situation even worse, the algorithms used to select the most stable reference genes failed to select the right reference genes. This was especially evident for the thorax tissue in *piceifrons*, as the only gene not exhibiting different expression levels between both rearing conditions, *RIBL5*, was ranked as the 8th most stable reference gene. Both geNorm and BestKeeper assume that perfect reference genes are strongly correlated to each other, and as a result they are susceptible to errors when the majority of the tested genes is actually influenced by the tested experimental conditions. Likewise, even though NormFinder takes both inter- and intra-group variation into account and should thus perform better under such conditions, this program ranked several genes showing a clear trend towards differential expression between rearing conditions as the most stable in *piceifrons*. Therefore, it is clearly not sufficient to only consider the overall stability values in selecting reference genes, but it is also important to take the actual $C_q$ values into account. Additionally, these findings suggest that we cannot simply use the same set of reference genes blindly across the species, but we must carefully consider the type of reference genes to be used for each species.

For all species, we followed the suggestions of the geNorm analysis for determining the amount of reference genes. For *piceifrons*, we thus suggest three genes should be used, while for the other species, two reference genes should be sufficient. In *piceifrons*, *Ann*, *Arm* and *Act5C* should not be selected as the reference genes when studying differential gene expression between isolated-reared and crowded-reared individuals. For such a study, we suggest the use of *RIBL5*, *Hsp70* and *EF2* for the head tissue, and *RIBL5*, *Hsp70* and *GAPDH* for the thorax tissue instead. Even though *RIBL5* was listed as the last-ranked reference gene by all three programs for the thorax tissue, we consider its inclusion is still valid for two reasons. First, it is the only gene that shows no differential expression between the density conditions. Second, the standard deviation on this gene, reported by BestKeeper, is also one of the lowest, even though this variable was not included in our ranking. After selecting the reference genes according to what is presented above to normalize gene expression of the four target genes, our results matched well with the results of RNA-seq experiments. For the other three species, the choice of reference genes is more straight-forward, as there was no discrepancy between the stability values and the $C_q$ values. Thus, we suggest the use of the two genes with the best overall stability ranking as reference genes in further studies. Our suggestions for each species are summarized in Table 7.

**Table 7  Suggestion for the use of reference genes in future RT-qPCR experiments for four *Schistocerca* species.**

| Species | Tissue | Suggested reference genes | | |
|---|---|---|---|---|
| *piceifrons* | head | RIBL5 | Hsp70 | EF2 |
| | thorax | RIBL5 | Hsp70 | GAPDH |
| *americana* | head | Act5C | EF2 | |
| | thorax | RIBL5 | EF2 | |
| *cubense* | head | Act5C | Hsp70 | |
| | thorax | Hsp70 | Ann | |
| *nitens* | head | Ann | Act5C | |
| | thorax | Ann | Arm | |

Two of the selected target genes for validation were pacifastins (*SPPP-4* and *SPPP-5*), which are members of a family of peptidase inhibitors that exhibit differential expression between isolated-reared and crowded-reared individuals in *S. gregaria* (*Badisco et al., 2011*; *Breugelmans et al., 2008*; *Breugelmans et al., 2009*; *Simonet et al., 2005*; *Simonet et al., 2004*). We show that *SPPP-4* expression in the thorax tissue from crowded-reared individuals is higher compared to isolated-reared individuals, and *SPPP-5* shows a similar but a non-significant trend, consistent with earlier studies in *S. gregaria* (*Badisco et al., 2011*; *Breugelmans et al., 2008*; *Simonet et al., 2005*). The other two target genes were allatotropin, a pleiotropic hormone that is best known as a stimulator of juvenile hormone (JH) production and a myostimulatory peptide (*Kataoka et al., 1989*; *Lismont et al., 2015*), and allatostatin, known to repress JH production in insects even though this function could not be confirmed in *S. gregaria* (*Stay, Tobe & Bendena, 1995*; *Verlinden et al., 2009*). We report that *ast* is significantly downregulated in crowded-reared locusts, and observed a similar but non-significant trend for *at*. We also note that these observations were not confirmed by our RNA sequencing data, where no significant differences were obtained for either of these two genes. Both *ast* and *at* exhibited very low expression levels, and the difference between both techniques might thus represent a difference in sensitivity between both techniques. Other differences between both technique (e.g., RNA sequencing being more prone to PCR artefacts and positional bias) might play an additional role in the observed differences. An earlier study in *S. gregaria* reported no significant difference between isolated-reared and crowded-reared individuals for *at* (*Lismont et al., 2015*), even though the data were not shown. Together, our data might suggest differences in JH regulation between isolated-reared and crowded-reared locusts.

## CONCLUSIONS

Our study is one of only few studies testing the stability of reference genes for qPCR in multiple species. When ranking reference genes based on their stability, we found clear differences between the four tested species. Additionally, all three used algorithms, geNorm, BestKeeper, and NormFinder, seemed to struggle when the majority of tested reference genes showed a trend towards differential expression in the tested experimental conditions. As such, stability values obtained by these algorithms should always be analyzed side by

side with the raw $C_q$ values. Additionally, our study suggests that both qPCR primer efficiency and reference gene stability should always be tested separately for each species of interest. Not doing so might result in differences in amplification efficiencies and incorrect interpretation of the results.

## ACKNOWLEDGEMENTS

We thank Mario Poot-Pech and the rest of the locust control team of the Comité Estatal de Sanidad Vegetal de Yucatán (CESVY), Mexico for their help in collecting *piceifrons* nymphs to establish our lab colony. We thank Ryan Selking, Charmelle Williams, and Sam Franklin for help with colony care, RNA extractions, and RT-qPCR. We thank Texas A&M AgriLife Genomics and Bioinformatics Service for data generation and Michael Dickens at Texas A&M High Performance Research Computing for installing all used tools into the Galaxy environment.

### Funding

This work was supported by the National Science Foundation (IOS-1253493). The funders had no role in study design, data collection and analysis, decision to publish, or preparation of the manuscript.

### Grant Disclosures

The following grant information was disclosed by the authors:
National Science Foundation: IOS-1253493.

### Competing Interests

The authors declare there are no competing interests.

### Author Contributions

- Bert Foquet conceived and designed the experiments, performed the experiments, analyzed the data, prepared figures and/or tables, authored or reviewed drafts of the paper, and approved the final draft.
- Hojun Song conceived and designed the experiments, prepared figures and/or tables, authored or reviewed drafts of the paper, and approved the final draft.

### Data Availability

Raw $C_q$ values for qPCRs are available in the Supplemental Files.

All raw sequencing reads are available at the NCBI SRA Bioproject: PRJNA633949.

Head and thorax transcriptome for *Schistocerca piceifrons* are available at NCBI TSA: GIOR00000000 and GIOS00000000.

### Supplemental Information

Supplemental information for this article can be found online at http://dx.doi.org/10.7717/peerj.9618#supplemental-information.

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
