# Peer review of "There is no magic bullet: the importance of testing reference gene stability in RT-qPCR experiments across multiple closely related species"

_PeerJ, doi:10.7717/peerj.9618_

## Round 0.1 · original submission · Minor Revisions

Thank you for submitting your manuscript to PeerJ. Based on the very positive comments of two reviewers I invite you to resubmit after making minor revisions. Both reviewers recognized the importance of this type of qPCR reference gene validation, especially in non-model species, and state that it can serve as a good example for future studies in other species. Please address all reviewer suggestions in your response letter with the resubmission

I have just a few suggestions to share:

1. It is difficult to see the paragraph breaks in the introduction and discussion. I would add lines between paragraphs or indent the first line. It would make reviewing the work easier.

2. The MIQE guidelines recommend using Cq in place of Ct. Is there a reason you decided to use Ct in this paper?

3. I would recommend including your R script for the T-test as a supplemental text file or provide a GitHub link to the script.

Reviewer 1 ·

Basic reporting

A few minor comments on the manuscript:

1) Line 45 is vague and needs more description.
2) Line 47 - are they referring to gene expression?
3) Line 167 - RNA extraction from insect tissue typically requires glass-bead disruption. It is not clear from methods whether this technique was used.
4) Line 232 - the authors state that they used three primer pairs for each gene. How did they determine which of the primer pairs to use in their analysis? In line 403, they indicate they used single primer pairs per each gene. This needs to be clarified in the methods.
5) Line 236 - For real-time reaction mixes, the authors indicate they used 5 ml of cDNA, 10 ml of SYBR green, and 5 ml of primers. This seems to be an error as it should be microliter quantities.
6) Table 2 - it would be helpful to indicate the size of PCR amplicons for each gene/species
7) Figure 2 - the figure legend is minimal and needs some more description.

Experimental design

No comment

Validity of the findings

Lines 327-344: The authors attempt to provide the rationale for why they selected 2 or 3 references genes based on stability. This is a confusing section and I suggest a re-write for better clarity and rationale.

Additional comments

In the manuscript, by Foquet and Song, the authors use a combination of RNA-sequencing data and quantitative real-time PCR to assess and validate the stability of reference genes used to quantify gene-expression changes in grasshoppers during the process of density-dependent phenotypic changes. This short report represents an important reference document for those studying gene expression changes in grasshoppers and can be used as a model for the determination of reference genes in other experimental systems. The manuscript is well-written and the analysis is comprehensive with appropriate literature references to support their findings. The experiments are well-designed and the data generated has been expertly analyzed. The presentation of the data in both figures and tables is sufficient. The conclusions are sufficient; however, a more clear description of the rationale for selecting reference genes would improve the manuscript.

Reviewer 2 ·

Basic reporting

The language is clear and concise, well referenced, and the figures are well-constructed and easy to understand. Tables 4 & 5 are a little confusing at first review, but I am at a loss as how to better organise them, as they are presenting (incorporating) multiple concepts and comparisons, which all need to be included. The RAW data appears to all be correct and support the figures and tables in the main text & figures. The hypotheses are quite clear and understandable.

Experimental design

Good paper in general (please see General Comments for suggested revisions & clarifications). Well constructed description of a neglected issue and topic, likely to have more than a trivial effect on many studies. Methods have been describe in proper detail, mostly.

Validity of the findings

This reviewer has encountered many of the same issues and concerns that the authors have presented, working in a non-model animal system(s). While there are some well-reviewed reference genes in model systems, in wild animals, or in non-traditional model systems, there are not well-documented reference genes. It is often difficult to include these unique reference genes in publications, however this paper strongly argues for the proper assessment of chosen reference genes and more importantly details why and how this might be done. Another important aspect of this paper is to point out the discrepancy in some of the software used to choose the "correct" reference gene(s). Well explained as to why they may not give the best result in specific situations.

Additional comments

Overall, a good, solid paper.

Suggestions:

(1) Why did the authors choose not to include the MIQE guidlines worksheet as a supplemental?

(2) Consider using ng*mL^(-1) as opposed to ng/ml

(3) Consider including specific catalogue numbers for some of the PCR reagents, especially the Taq enzyme. Was the Taq enzyme a hot start (HS) variant? If not, why not?

(4) Please consider a brief discussion on the nature of the discrepancies between RNA-Seq & RT-qPCR data results. Because these are two totally different techniques as described in your paper it is unlikely that they would give similar results. RNA-Seq without UMIs is prone to significant PCR artefacts that RT-qPCR is not, and RNA-Seq has positional effects (3' bias) that RT-qPCR may or may not have depending on the positioning (choice) of the qPCR primers within the gene/exon.

Minor corrections:

(a) Was a value of 3.9 RQN considered "good"? Is that scale from 0 to 10? If so, for locusts, are the scores skewed because of the nature of the totalRNA? Perhaps an explanation would be warranted?

(b) Why were the reads "converted" to Sanger format? The CASAVA 1.8 pipepline (by Illumina) onwards uses a standard Phred score system that is compatible with Sanger - unless I'm mistaken.

(c) What were the contaminating sequences? (line 191)

(d) What was the nature of the in silico normalisation method used (line 193)

(e) Lines 236-237, I believe that the mL should be microL (the "m" symbol got incorrectly encoded).

(f) Were there any primer-dimers as seen by agarose gel analysis (line 287-288).

(g) Line 362-363: "qPCR results agreed well ..." Not really, see point #4 above.

(h) Figure 2: The colouration of the bars is difficult to discern with the black being very close to the dark grey on a printout - can these be maybe switched to patterns (hashes/dots/etc).

(i) Table 4 & 5: You took the geometric mean, did you consider "voting" to choose the best gene amongst the three different software evaluations.

---

## Round 0.2 · accepted · Accept

Thank you for resubmitting your manuscript and your work to address the comments of the reviewers. The two original reviewers are satisfied that you have addressed their suggestions and I am happy to now accept your paper for publication in PeerJ. Reviewer 2 has a minor request about your supplemental R script that you could address in the final draft.

You will be given the option to make the reviews of your manuscript available to readers. Please consider doing so as this review record can be a great resource for readers of your paper and contributes to more transparent science.

Thank you for choosing PeerJ as a venue for publishing your work.

Reviewer 1 ·

Basic reporting

No Comment

Experimental design

No Comment

Validity of the findings

No Comment

Additional comments

Thank you for addressing the concerns and suggestions of the reviewer.

Reviewer 2 ·

Basic reporting

Good

Experimental design

Good

Validity of the findings

Good

Additional comments

Thank you for the corrections. Please include the version of R that the script was written for in the remarks (REM Statement) of the script, so that in case subsequent iterations render the script non-functional, one can benchmark the code.

Other than that, manuscript is ready-to-go!